# Late killing of *Plasmodium berghei* sporozoites in the liver by an anti-circumsporozoite protein antibody

**Manuela C Aguirre-Botero[1], Olga Pacios[1†], Susanna Celli[1], Eduardo Aliprandini[1‡], Anisha Gladston[1,2], Jean-Michel Thiberge[1], Pauline Formaglio[1§], Rogerio Amino[1*]**

[1]Institut Pasteur, Université Paris Cité, Malaria Infection and Immunity, BioSPC, Paris, France; [2]Department of Life Sciences, Imperial College London, London, United Kingdom

**\*For correspondence:**
rogerio.amino@pasteur.fr

**Present address:** [†]Microbiology Department - A Coruña Hospital, MicroTM-INIBIC, A Coruña, Spain; [‡]Affinity Reagents Team, Biologics Engineering, AstraZeneca, Cambridge, United Kingdom; [§]Sorbonne Université, Inserm U1135, CNRS ERL8255, Centre d'Immunologie et des Maladies Infectieuses (CIMI-Paris), 91 10 Boulevard de l'Hôpital, Paris, France

## eLife Assessment

This **important** study shows that a monoclonal antibody against the repetitive region of the circumsporozoite protein (CSP) of the Malaria-causing parasite P. berghei has neutralizing activity on parasite invasion and development. The authors present **convincing** in vivo data confirming previous in vitro work, that suggested the intracellular post -invasion effect for this antibody. The findings offer insights into the inhibitory action of this anti-CSP antibody, which could inform the development of more effective malaria vaccines and therapeutic antibodies. "
[Editors' note: this paper was reviewed by Review Commons.]

**Abstract** *Plasmodium* sporozoites are inoculated into the skin during the bite of an infected mosquito. This motile stage invades cutaneous blood vessels to reach the liver and infect hepatocytes. The circumsporozoite protein (CSP) on the sporozoite surface is an important antigen targeted by protective antibodies (Abs) in immunoprophylaxis or elicited by vaccination. Antibody-mediated protection mainly unfolds during parasite skin migration, but rare and potent protective Abs additionally neutralize sporozoite in the liver. Here, using a rodent malaria model, microscopy and bioluminescence imaging, we show a late-neutralizing effect of 3D11 anti-CSP monoclonal antibody (mAb) in the liver. The need for several hours to eliminate parasites in the liver was associated with an accumulation of 3D11 effects, starting with the inhibition of sporozoite motility, sinusoidal extravasation, cell invasion, and terminating with the parasite killing inside the invaded cell. This late-neutralizing activity could be helpful to identify more potent therapeutic mAbs with stronger activity in the liver.

## Introduction

Malaria remains the deadliest parasitic disease worldwide and is responsible every year for over 600,000 deaths, mainly in sub-Saharan Africa (**WHO, 2023**). The infection, caused by parasites of the genus *Plasmodium*, begins with the inoculation of the sporozoite stage into the skin by an infected mosquito (**Sidjanski and Vanderberg, 1997**). Within the skin, sporozoites become activated and move through the dermis until some find and invade a blood vessel (**Amino et al., 2006**; **Vanderberg and Frevert, 2004**). Upon entering circulation, parasites arrest in the liver (**Tavares et al., 2017**), cross the sinusoidal barrier within 2–3 hr (**Tavares et al., 2013**), and eventually invade hepatocytes. Intracellular parasites replicate massively before being released back into the blood to infect erythrocytes and cause the disease (**Sturm et al., 2006**; **Tarun et al., 2006**). The sporozoite surface is covered

by a dense coat of the circumsporozoite protein (CSP), shown to be an immunodominant protective antigen in a rodent malaria model (*Zavala et al., 1983*). This protein has a highly conserved structural organization across diverse *Plasmodium* species, comprising conserved N- and C-terminal domains and a highly variable and repetitive central region. Due to its surface localization, abundance, conservation between the same species and immunodominant character, CSP has been targeted by the most effective malaria vaccines. For example, RTS,S/AS01 and R21/Matrix M, the only two malaria vaccines recommended by the World Health Organization (WHO), include a fragment of the central repetitive region and the C-terminal domain of the CSP from *Plasmodium falciparum* (Pf), the most lethal human-infecting parasite (*de Almeida et al., 2021*). Extensive research has demonstrated a positive correlation between high levels of anti-CSP antibodies (Abs) induced by the RTS,S/AS01 vaccine and efficacy against malaria (*White et al., 2013*; *White et al., 2014*; *White et al., 2015*). Remarkably, anti-CSP monoclonal Abs (mAbs) protect in vivo against sporozoite infection in various experimental settings, including mice (*Imkeller et al., 2018*; *Kisalu et al., 2018*; *Murugan et al., 2018*; *Oyen et al., 2017*; *Potocnjak et al., 1980*; *Tan et al., 2018*; *Wang et al., 2020*; *Yoshida et al., 1980*), monkeys (*Cochrane et al., 1982*), and humans (*Gaudinski et al., 2021*; *Kayentao et al., 2022*; *Wu et al., 2022*).

It has been observed that Abs that target the CSP central repetitive region exhibit a cytotoxic effect on the sporozoites, accompanied by the shedding of the CSP surface coat (*Stewart and Vanderberg, 1991*; *Yoshida et al., 1980*), inhibition of motility, loss of infectivity (*Hollingdale et al., 1982*; *Stewart et al., 1986*), and parasite death (*Aguirre-Botero et al., 2023*; *Aliprandini et al., 2018*). These effects are independent of downstream host immune effectors such as the complement and immune host cells (*Aliprandini et al., 2018*). Instead, Ab cytotoxicity increases when parasites are moving in 3D substrates such as the matrix of the cutaneous environment (*Aguirre-Botero et al., 2023*). Accordingly, neutralization of sporozoites by anti-CSP Abs occurs primarily during their migration through the skin (*Aguirre-Botero et al., 2023*; *Aliprandini et al., 2018*) and is dependent on sporozoite motility and host cell-wounding activities (*Aliprandini et al., 2018*), which are both crucial for parasite progression in the dermis (*Amino et al., 2008*).

Yet, while the role of the skin in anti-CSP Abs-mediated sporozoite neutralization has been increasingly studied, little is known about the mechanisms by which these effectors contribute, outside of the skin, to neutralize sporozoite infection. Potent Abs or high concentrations of them were shown to efficiently target sporozoites in the blood and liver (BL) following intravenous (i.v.) parasite inoculation which bypasses the skin (*Aguirre-Botero et al., 2023*; *Potocnjak et al., 1980*; *Wang et al., 2020*). Ab cytotoxicity is also thought to be responsible for sporozoite neutralization in the liver (*Wang et al., 2020*), but some highly cytotoxic Abs were not able to significantly decrease infection following an i.v. sporozoite inoculation (*Aguirre-Botero et al., 2023*). In this study, we analyzed how 3D11, a cytotoxic mAb against the CSP of *Plasmodium berghei* (Pb) which has been shown to inhibit parasite motility and kill sporozoites in vitro (*Aliprandini et al., 2018*), as well as to inhibit motility in the skin (*Flores-Garcia et al., 2018*), affects sporozoite distribution in the BL, hepatocyte infection, and intracellular development. The current investigation provides a comprehensive view of the kinetics and mode of action of 3D11 on multiple steps of liver infection.

## Results

### Tissue-dependent neutralization of Pb sporozoites

To assess how efficiently the anti-PbCSP repeat mAb 3D11 targets sporozoites in the skin and in the BL, mice were passively immunized intraperitoneally with 100 µg of mAb 3D11 or with an isotype control (W24). Twenty-four hours later, animals were challenged either i.v. or micro-injected into the skin with, respectively, 1000 and 5000 GFP-expressing sporozoites. The sporozoite inoculum was adjusted to achieve comparable infection regardless of the route of parasite injection as reflected by the similar parasitemia of the control groups, measured by the percentage of infected red blood cells by flow cytometry (*Figure 1A*, W24). In addition, we compared the results obtained following these two routes of inoculation with infection initiated by sporozoites naturally transmitted by mosquitos. The extent of bite exposure was adjusted to reach a level of host infection equivalent to that obtained after skin and i.v. challenge. Protection was assessed by measuring the parasitemia 5 days after sporozoite inoculation when parasites are still growing exponentially in the blood (*Figure 1A*). Sterile protection

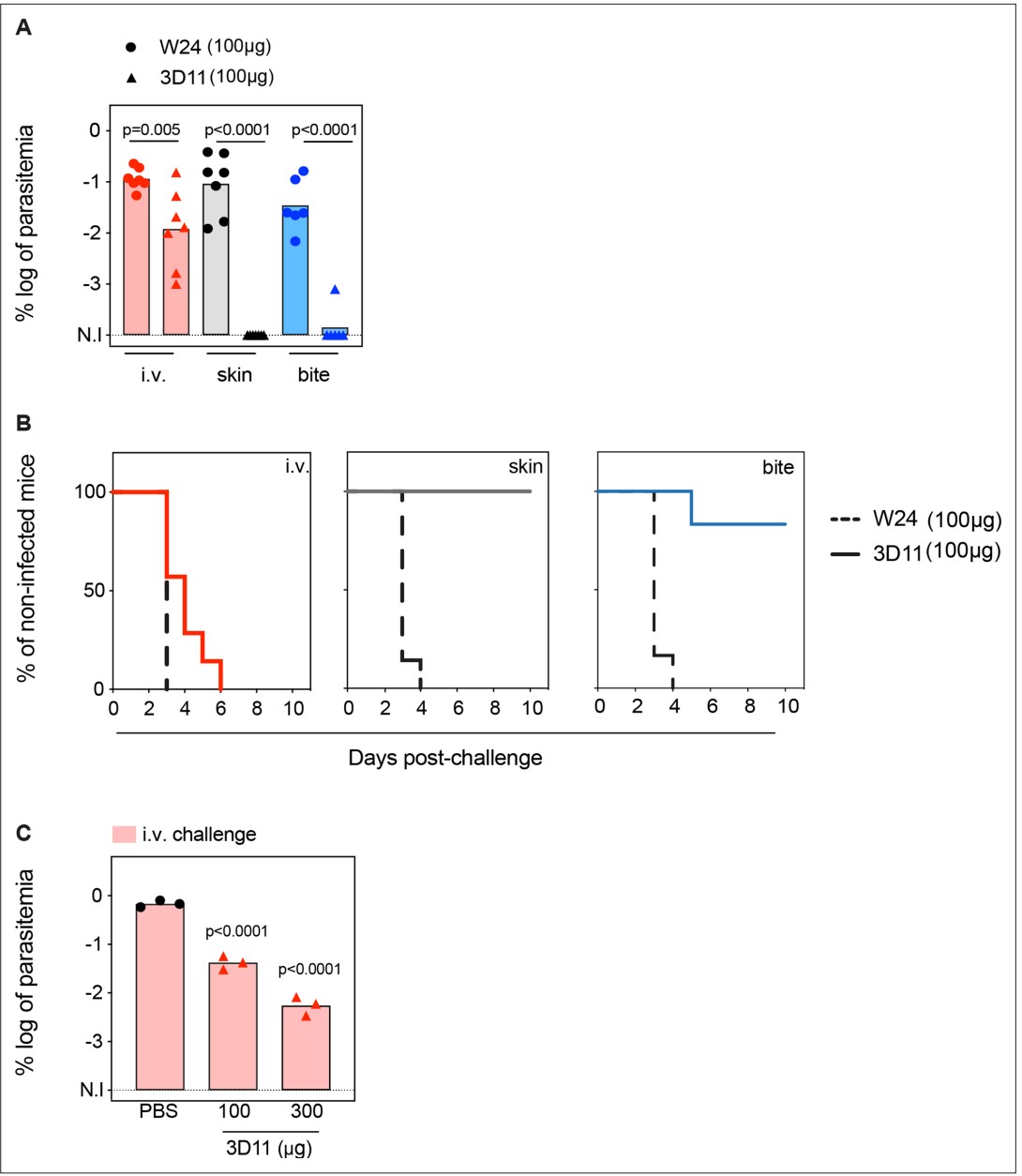

**Figure 1.** Tissue-dependent neutralization of Pb sporozoites. (**A**) Comparison of the log of percentual parasitemia at day 5 post-challenge and (**B**) survival curves of animals that received 100 μg of the isotype control (W24) or 3D11 monoclonal antibody (mAb), followed by inoculation of sporozoites, in the tail vein (i.v.), micro-injected in the skin or by mosquito bite (bite). For i.v. and skin n = 7 mice. For mosquito bite n = 6 mice. N.I: non-infected. (**C**) Titration of the protection elicited by 3D11 after the i.v. challenge, 24 hr after intraperitoneal transfer of the antibody (n = 3). (**A, C**) Statistical significance was determined by one-way ANOVA with Holm–Šídák correction for multiple comparisons.

was defined as the percentage of uninfected animals 10 days after the challenge (*Figure 1B*). The skin and the bite challenge allowed us to study 3D11 neutralizing activity in the cutaneous tissue as well as in the BL, while the i.v. challenge bypasses the skin, and reflects only the mAb-neutralizing activity in the BL.

Passive transfer of 100 μg of mAb 3D11 sterilely protected animals only when sporozoites were inoculated in the skin, either by micro-injection (100%) or by mosquito bite (86%, *Figure 1A, B*). Despite not eliciting sterile protection, 3D11 led to a tenfold reduction in parasitemia after i.v. challenge (*Figure 1A*), and accordingly, a delayed prepatency in passively immunized mice compared to

the control group (*Figure 1B*). These results confirm that the bite challenge is more comparable to the skin challenge than the i.v. challenge, as previously observed (*Aguirre-Botero et al., 2023*; *Aliprandini et al., 2018*). Furthermore, we confirmed that 3D11 predominantly neutralizes sporozoites in the skin, but also targets sporozoites in the BL (*Flores-Garcia et al., 2018*), establishing a quantitative measure of its relative tissue protection.

Next, we investigated the impact of higher mAb concentration on sporozoite neutralization in the BL. Following the same passive immunization schedule, we compared the parasitemia of mice injected intraperitoneally with 100 and 300 µg of mAb 3D11 and challenged i.v. 24 hr later. Increasing the mAb dose to 300 µg further reduced the parasitemia by ~100-fold compared to the control (*Figure 1C*). However, this threefold increase of the mAb dose still failed to confer sterile protection against i.v. challenge while 100 µg of 3D11 were sufficient to clear the infection in 86–100% of the mice after skin or bite challenge (*Figure 1A, C*). These results indicate that 3D11 sporozoite neutralization in the BL is concentration dependent but is not as efficient as in the skin.

## Kinetics of Pb sporozoite elimination in the liver

To further characterize 3D11 sporozoite neutralization in the BL, we first quantified the biodistribution and elimination of parasites in the mouse body by bioluminescence imaging following infection with sporozoites constitutively expressing a GFP-luciferase fusion protein. By comparing parasite load in 3D11- and PBS-treated mice at four critical time points covering the parasite hepatic life cycle (*Figure 2A*), we measured the potential of 3D11 to (1) kill parasites in the blood, (2) inhibit the specific arrest of sporozoites in the liver, and (3) hinder their intracellular development.

For the quantification of parasite load in the hepatic tissue, the body of each animal was divided into four regions of interest (ROIs), and the bioluminescent signal emanating from the upper abdominal ROI was attributed to the liver (*Figure 2B*; *Tavares et al., 2017*). Mice were i.v. injected with PBS or 100 µg of 3D11 to maximize mAb transfer into the blood circulation. Bioluminescence distribution in the mouse body was then measured 7 min after parasite inoculation to quantify the specific arrest of sporozoites in the liver (*Tavares et al., 2017*). By calculating the ratio between the total flux of photons in the hepatic ROI and the total flux in the whole body over 5 min acquisition (*Figure 2B*), we estimated the percentage of parasites accumulating in the liver between 7 and 12 min after sporozoite injection (*Figure 2B*, left: control, right: 3D11-treated mouse). The total flux and the percentage of signal in the liver were not affected by the presence of 3D11 (83% for control vs. 87% for 3D11-treated mice p = 0.36, *Figure 2C, D*, 7 min), indicating that 3D11 neither significantly killed parasites at this early time point of infection nor affected sporozoite arrest in the liver.

Bioluminescence was further measured in the same animals 4, 24, 44 hr (*Figure 2*), and 72 hr later (*Figure 2—figure supplement 1*). Since most sporozoites cross the liver sinusoidal barrier within 3 hr post-i.v. inoculation (*Mathieu et al., 2015*; *Tavares et al., 2013*), the bioluminescent signal acquired in the hepatic region at 4 hr in the control group should correspond to sporozoites inside the liver parenchyma, while between 24 and 44 hr, the signal should reflect the development of exoerythrocytic forms (EEFs) within the hepatocytes (*Figure 2A*). At 4 hr, the difference between the 3D11- and the vehicle-treated group was ~twofold, but it was not statistically different. In comparison, at 24 hr, a >20-fold decrease in the relative parasite load was evident (p = 0.008, *Figure 2D, E*). This difference slightly increased but not significantly from 24 to 72 hr (*Figure 2D*, *Figure 2—figure supplement 1*), suggesting a potential although small effect of 3D11 mAb at this concentration on late liver stages. Altogether, these data indicate that 3D11, rather than having an early effect on i.v. inoculated sporozoites in the blood circulation for example, by inhibiting the homing or killing the parasite in the blood, requires more than 4 hr to eliminate most parasites in the liver.

## The effect of a cytotoxic mAb on sporozoite crossing of the sinusoidal barrier

In vivo bioluminescence imaging of liver infection showed that 3D11 mAb eliminated most of the parasites between 4 and 24 hr post-i.v. inoculation. Yet, it remained unclear whether parasites were dying in the lumen of the hepatic sinusoids or inside the hepatic parenchyma. To better observe 3D11's neutralizing mode of action, we imaged parasites in the liver using intravital fluorescence high-speed confocal microscopy.

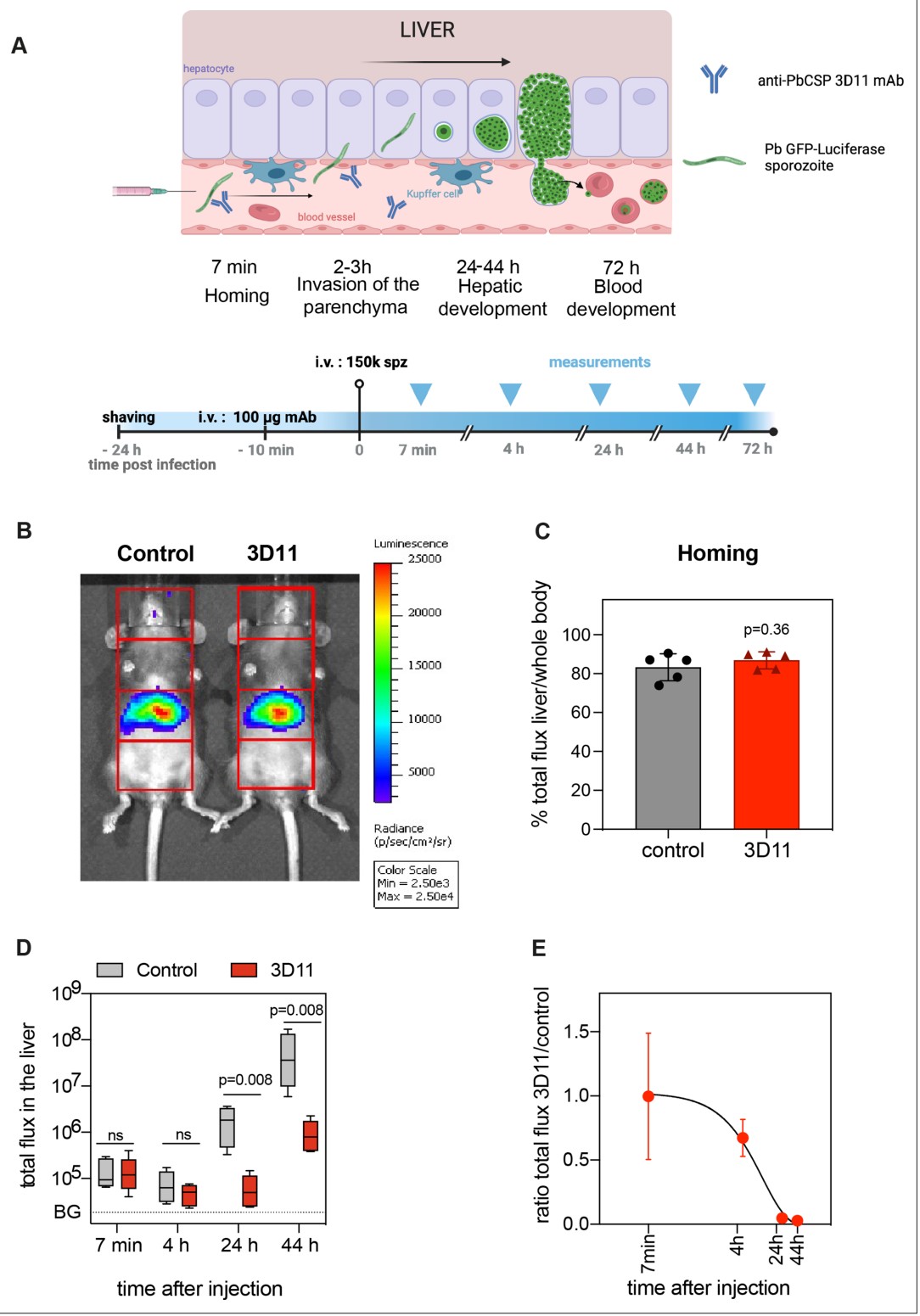

**Figure 2.** Kinetics of Pb sporozoite elimination in the liver. (**A–E**) To assess the kinetics of parasite killing by 3D11, either PBS (control) or 100 µg of 3D11 mAb (3D11) were i.v. injected in mice 10–15 min before the i.v. inoculation of sporozoites expressing GFP-luciferase. Parasite load in the liver was assessed by bioluminescence. (**A**) Top: Schematic representation of sporozoite homing, invasion, and development in the liver after i.v. injection. Bottom: Time schedule of the experiment. spz: sporozoites. (**B**) Exemplary picture of the recorded bioluminescence images and selected regions of interest (in red, ROIs). The signal in the third quadrant was used to measure liver infection.

*Figure 2 continued on next page*

*Figure 2 continued*

Measurement was performed 7 min after spz inoculation in control and mice passively transferred with 3D11. Scale in radiance (photons/s/cm²/sr). (**C**) Quantification of the percentage of the signal collected from the liver compared to the whole body. Statistical significance was analyzed using the unpaired *t*-test. (**D**) Total flux (photons/s) in the liver of mice from control and 3D11 groups. Statistical significance was determined by Kruskal–Wallis test. BG: background signal. (**E**) The ratio of the total flux in the liver between control and 3D11mAb-treated mice. (**C–E**) *n* = 5 mice from two independent experiments. Data are presented as means ± SD. The cartoon was created with BioRender.com.

The online version of this article includes the following figure supplement(s) for figure 2:

**Figure supplement 1.** Comparison of the ratio of total flux between 3D11-treated and control mice 24, 44, and 72 hr post-infection (*n* = 3 animals).

A titanium window was surgically inserted in the abdomen of mice (*Ritsma et al., 2013*) expressing GFP in endothelial cells (*Xu et al., 2010*), allowing for the longitudinal visualization of sporozoites and liver sinusoids. For these experiments, we followed the same experimental design as for the bioluminescent assay (*Figure 3A*). After the i.v. injection of 100 µg of 3D11, mice were anesthetized and placed on the heated stage of an inverted microscope. Similar amounts of 3D11-susceptible Pb sporozoites expressing mCherry (Pb mCherry) and non-susceptible Pb sporozoites expressing PfCSP and GFP (PbPf GFP) were then co-injected (*Figure 3—figure supplement 1A*) while simultaneously recording to capture the first moments of parasite arrest in the liver sinusoids (*Tavares et al., 2017*). Equivalent numbers of susceptible (*n* = 80) and non-susceptible (*n* = 86) sporozoites were observed by microscopy in the analyzed fields, indicating that 3D11 did not impair parasite arrest in the liver sinusoids or significantly kill sporozoites during the first 2 hr after i.v. inoculation (*Figure 3B*), as observed by bioluminescence (*Figure 2C*, 7 min).

Next, we quantified the inhibition of sporozoite motility by 3D11 in vivo. Sporozoites were scored as motile if they moved at least one body length in 5 min, otherwise, they were considered non-motile. Additionally, we also quantified if the parasites were inside the vessel or outside that is, in the hepatic parenchyma (*Figure 3C*, *Figure 3—figure supplement 2*). Since 3D11 does not target PbPf GFP parasites, most of them are motile in the first hour of analysis, making them easily distinguishable from GFP-expressing endothelial cells (*Figure 3—figure supplement 2*). Already 10 min after inoculation, 3D11-susceptible parasites displayed a strong impairment in motility compared to the control parasites in passively immunized mice (*Figure 3B*). At this time point, sporozoites were only occasionally observed dying during these first minutes. Thirty minutes after sporozoite injection, almost none of the susceptible Pb mCherry parasites were motile while 50% of the control PbPf GFP sporozoites were still moving. Two hours later, a similar number of PbPf (*n* = 18, *Figure 3D*) and Pb (*n* = 29, *Figure 3D*) sporozoites were found in the liver, all, already immotile. Strikingly, only 52% of the Pb mCherry susceptible sporozoites were found in the parenchyma in contrast to 89% of non-susceptible parasites, indicating that 3D11 affects sporozoite entry into the hepatic parenchyma (*Figure 3D*). To exclude any effect coming from the insertion of the window on these results, additional experiments were performed by imaging dissected livers of passively immunized mice. Ex vivo, we observed that after 1 hr, 97% of the control parasites (*n* = 34) were inside the parenchyma in contrast to 53% of the susceptible sporozoites (*n* = 45), confirming the results of the in vivo experiments. Since comparable numbers of Pb mCherry and PbPf GFP sporozoites were found in the microscopic fields quantified in vivo and ex vivo (*Figure 3D*), we can infer that 3D11 blocks sporozoite entry into the hepatic parenchyma by around twofold. Yet, this impairment alone cannot explain the decrease of at least one order of magnitude in the parasitemia measured 3–4 days after infection in the mice used for in vivo imaging (*Figure 3—figure supplement 1B*). Hence, in 3D11-treated mice, most sporozoites die in the hepatic parenchyma probably due to the loss of fitness caused by the 3D11 mAb. However, whether the late reduction in parasite load occurring after 4 hr post-infection is associated with the death of extracellular sporozoites that failed to invade hepatocytes or corresponds to the clearance of parasites undergoing abortive intracellular development could not be addressed in vivo.

## In vitro effect of 3D11 on sporozoite invasion and development

To better understand the late-neutralizing effect of 3D11 mAb observed on sporozoites in vivo, we performed an invasion and development assay in vitro (*Prudêncio et al., 2008*) using Pb GFP-expressing

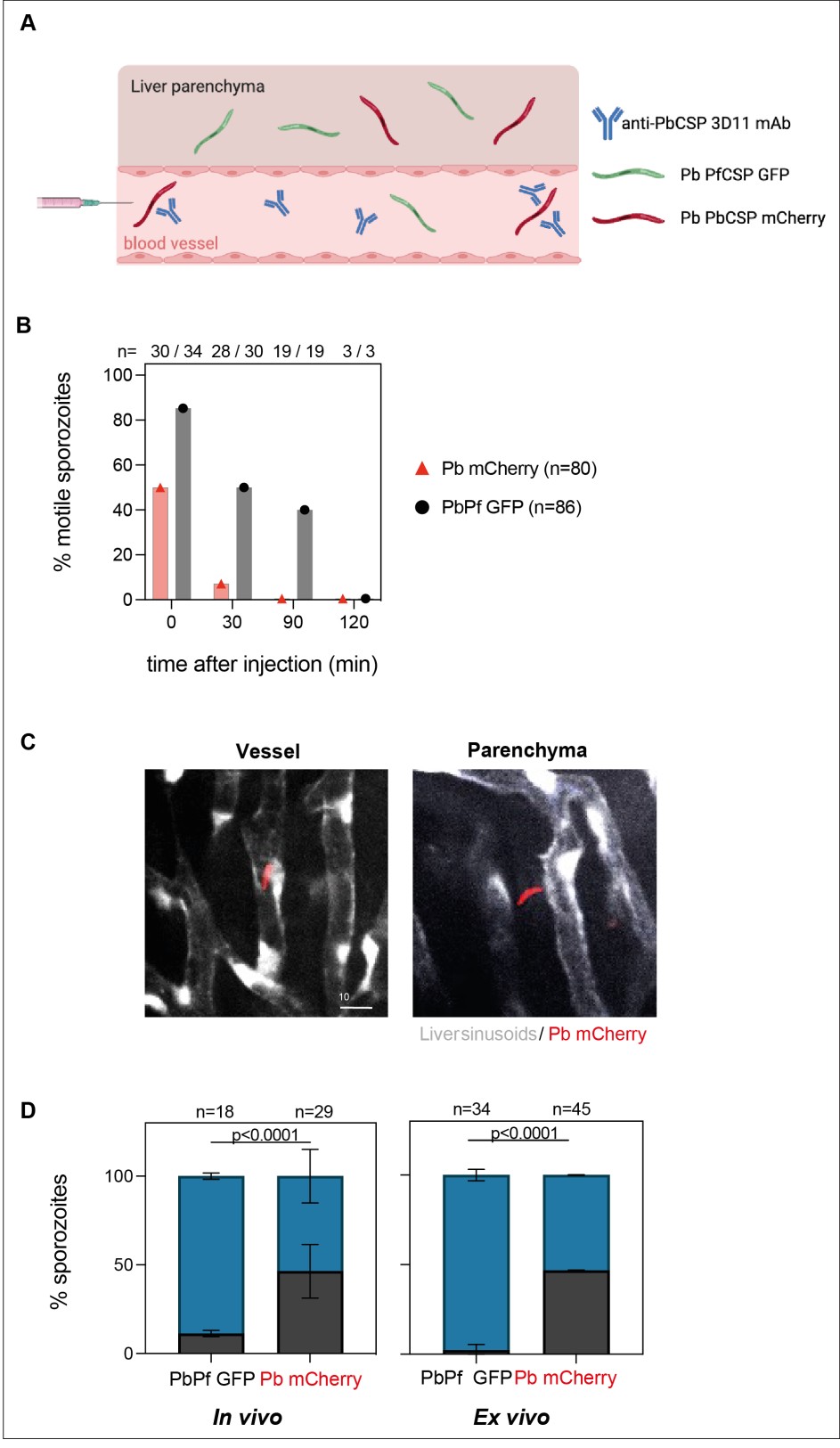

**Figure 3.** 3D11 inhibits sporozoite motility and partially blocks entry into the liver parenchyma. (**A**) Graphical representation of the experimental setup to analyze sporozoite localization in the liver. For intravital experiments (in vivo), flk1-GFP female mice were transferred intravenously with 100 µg of 3D11. Ten minutes later, the animals were placed on the microscope stage and susceptible (Pb mCherry) and control sporozoites (PbPf GFP) were co-

*Figure 3 continued on next page*

*Figure 3 continued*

inoculated i.v. into the tail vein. For ex vivo experiments, 3D11 was transferred 10 min before sporozoite injection, and 1 hr later the liver was dissected and imaged. (**B**) Quantification of motility inhibition over time. Data from four independent experiments. (**C**) Representative images of one focal plane showing sporozoites inside the sinusoid lumen (left panel) or the hepatic parenchyma (right panel). Pb mCherry is shown in red and the GFP expressed by liver sinusoidal endothelial cells in white. Scale bar: 10μm. (**D**) Left panel: Quantification of the localization of Pb mCherry (red) and PbPf GFP (black) sporozoites determined in vivo between 30 min to 3 hr post-inoculation. Data from two independent experiments including a total of three animals. Statistical significance was determined by the Chi-square test. Right panel: Quantification of sporozoite localization ex vivo determined 1 hr post-inoculation. Data from two independent experiments. On top of the graphs is represented the number of quantified sporozoites. Data are presented as means ± SD. Cartoon was done with BioRender.com.

The online version of this article includes the following figure supplement(s) for figure 3:

**Figure supplement 1.** Experimental settings and additional results related to *Figure 3*.

**Figure supplement 2.** Representative images of one focal plane showing PbPf GFP sporozoites inside the vessel or the hepatic parenchyma.

sporozoites to infect the hepatoma cell line HepG2 in the presence or absence of 3D11. We quantified sporozoite invasion at 2 hr and ensuing intracellular development at 15 or 44 hr post-infection by flow cytometry. To overcome the variable invasion rates achieved across multiple experiments, the number of GFP+ cells in the cultures treated with 3D11 was normalized by the corresponding value in matching PBS-treated cultures and expressed as a percentage of control. In addition, by comparing the total number of recovered GFP+ sporozoites at 2 hr in the two studied conditions, we measured the early lethality (%viable sporozoites, *Figure 4B*) of the anti-CSP Ab on the extracellular forms of the parasite (*Figure 4A*).

At 2 hr, increasing concentrations of 3D11 were associated with a drastic reduction in the number of recovered extracellular sporozoites as well as a marked decrease in the percentage of GFP+ invaded cells compared to PBS controls. 3D11, therefore, kills and inhibits parasite entry into cells in a dose-dependent manner (*Figure 4B*, 2 hr). The inhibition patterns of cell invasion and parasite killing were similar suggesting that the Ab's cytotoxic effect could account for part of the invasion inhibition in this setting. However, the causality of this association still needs to be demonstrated since at low 3D11 concentrations the inhibition of invasion was higher than killing.

To determine if the cytotoxic effect of 3D11 also causes delayed parasite killing within HepG2 cells, we compared the percentage of surviving and developing parasites relative to the control at 2, 15, and 44 hr (*Figure 4C*). Interestingly, we observed a dose-dependent decrease in the number of intracellularly developing parasites when comparing the recently invaded cells (2 hr) with later time points (*Figure 4C*, 2 hr vs. 15 and 44 hr). Notably, this effect was already reached at 15 hr. For example, the exposition of sporozoites for 2 hr with 5 μg/ml of 3D11 led to a more than 50% reduction in the post-invasion number of intracellular parasites at 15 and 44 hr (*Figure 4D*). The dose-dependent loss of fitness caused by 3D11 mAb was also demonstrated by decreasing GFP intensities of the surviving EEFs (*Figure 4E*). Post-invasion labeling of 3D11 bound to the membrane of intracellular parasites revealed a strong staining surrounding the parasite at 2 and 15 hr, but only punctual traces of 3D11 at 44 hr (*Figure 4F*, 3D11 - cyan). Of note, CSP was detected surrounding the control parasites at all time points indicating that the lack of staining at 44 hr is not due to a decrease in the CSP amount on the parasite surface (*Figure 4F*, CSP, Control). To evaluate the potential post-invasion entry of 3D11 into the PV of infected cells and posterior neutralization of intracellular parasites, we incubated invaded cells from 2 to 44 hr with 3D11, but no effect on the parasite intracellular development was observed (*Figure 4G*, 2 hr p.i.). 3D11 incubated for 2 hr with sporozoites and cells elicited, as expected, a dose-dependent inhibition of parasite development. Altogether, our results indicate that the late inhibition of parasite development is already achieved at 15 hr and likely caused by Abs dragged inside cells bound to sporozoites before or during the invasion. In line with the late sporozoite neutralization effect observed in the liver, the in vitro assay demonstrates that 3D11 not only inhibits sporozoite entry into host cells but also affects its fitness resulting in subsequent parasite killing hours after invasion.

To further characterize the killing of intracellular parasites by 3D11 in HepG2 cells, we next evaluated the expression of the parasitophorous vacuole membrane (PVM) marker, UIS4 (*Mueller et al.,*

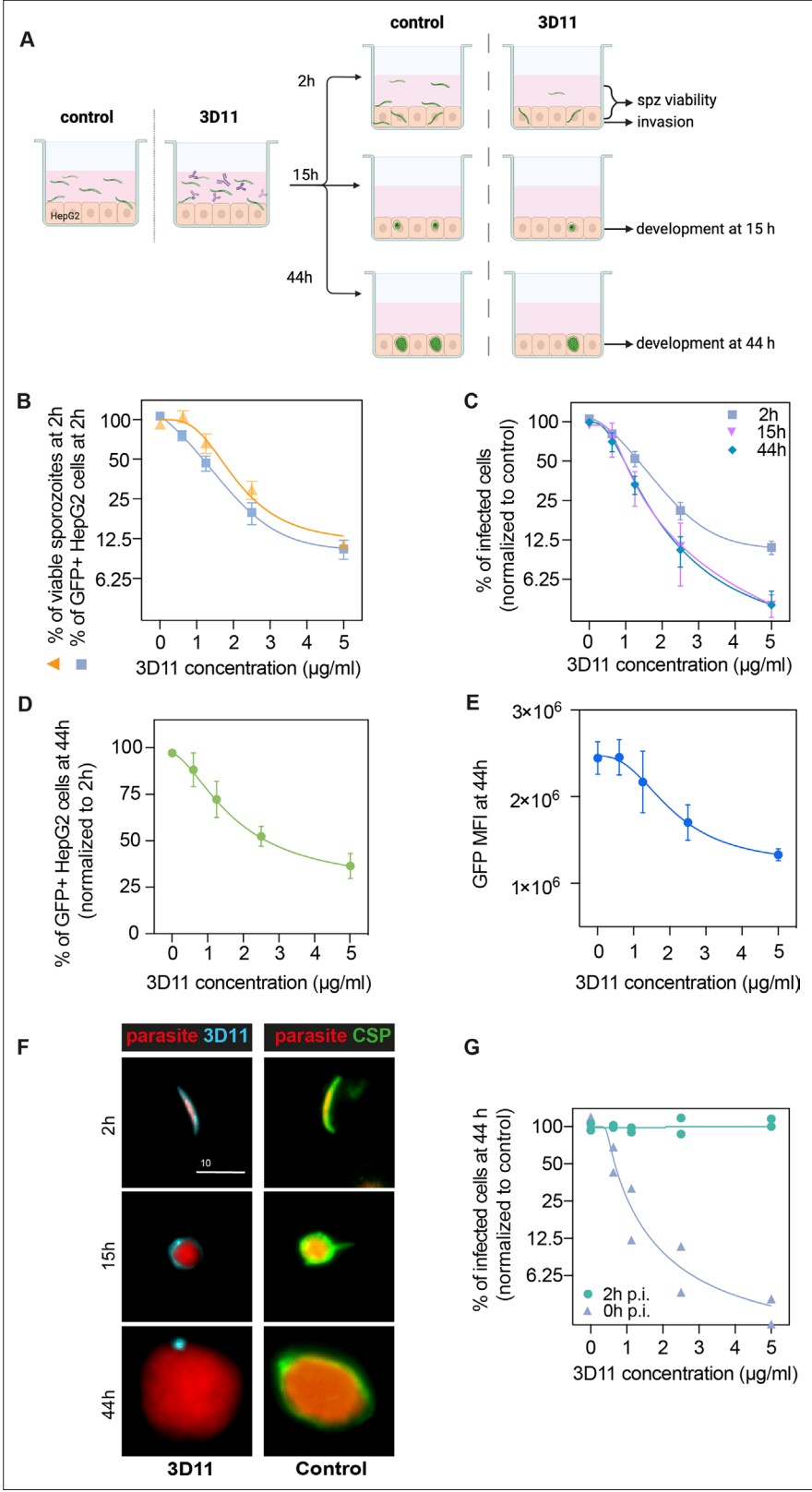

**Figure 4.** 3D11 inhibits sporozoite invasion and development in vitro. (**A**) Experimental setup to measure the impact of 3D11 on sporozoite invasion and exoerythrocytic form (EEF) development in HepG2 cells. Each experiment was performed in triplicates in the presence or absence of different concentrations of 3D11. Two identical plates were prepared to assess invasion at 2 hr and development at 15 and 44 hr post-infection. Cells,

*Figure 4 continued on next page*

*Figure 4 continued*

as well as the supernatant, were analyzed by flow cytometry (**B**). Inhibition of HepG2 invasion by sporozoites was measured 2 hr after parasites were added to the culture in the presence of 3D11 (blue square). In the same assay, sporozoite viability was quantified as the percentage of viable sporozoites after 2 hr relative to the control (yellow triangle). (**C**) In addition to the invasion assay, development inhibition was quantified at 15 (pink inverted triangle) and 44 hr (blue rhombus) after sporozoite addition to the cells. (**D**) Comparison between the percentage of infected HepG2 cells at 44 and 2 hr. (**E**) Mean fluorescent intensity (MFI) of EEFs 44 hr post-infection. (**B–E**) Data were combined from three to seven independent experiments. Bars represent the mean ± SEM. (**F**) Left: Representative images of intracellular parasites after being treated with 1.25 µg/ml 3D11. For treated parasites: 2, 15, and 44 hr after sporozoite and 3D11 addition, the cultures were washed, fixed, and permeabilized, allowing the internalized 3D11 to be revealed using a goat anti-mouse 647 antibody (cyan). Right: Control parasites were labeled post-permeabilization with 3D11 to visualize circumsporozoite protein (CSP) expression (green). Scale bar: 10µm. (**G**) The effect of 3D11 after sporozoite invasion was measured by adding 3D11 2 hr after sporozoites were added to the culture (purple triangle, 2 hr). 3D11 was added simultaneously with the sporozoites for 2 hr in the positive control group (green circle, 0 hr). Symbols represent two independent experiments, each with two technical replicates. Schematic created using BioRender.com.

The online version of this article includes the following figure supplement(s) for figure 4:

**Figure supplement 1.** Gating strategy to analyze parasite mortality, invasion, and development in the absence or presence of 3D11.

*2005*), to infer the parasite intracellular development at 2, 4, and 44 hr. HepG2 cells were incubated with Pb-GFP-expressing sporozoites in the absence (Control, *Figure 5*) or presence of 1.25 µg/ml of 3D11 during the first 2 hr of incubation (3D11, *Figure 5*). The chosen 3D11 concentration led to ~50% decrease in cell invasion (*Figure 4C*, 2 hr) and ~30% decrease in the post-invasion number of EEFs (*Figure 4D*), leaving enough parasites to be analyzed by microscopy. To distinguish between extracellular and intracellular parasites at 2 hr, washed and fixed samples were incubated with mouse 3D11 mAb (1 µg/ml) and revealed with a fluorescent anti-mouse secondary antibody (*Figure 5A*, 3D11 in blue). Samples were then permeabilized and incubated with a goat anti-UIS4 polyclonal antibody revealed with a fluorescent anti-goat secondary antibody (*Figure 5A*, UIS4 in red). DNA was stained with Hoechst (*Figure 5A*, DNA in white).

Extracellular GFP$^+$ sporozoites were identified by their 3D11$^+$UIS4$^-$ phenotype (*Figure 5A*, 2 hr, extracellular). Conversely, intracellular parasites were identified by their 3D11$^-$ phenotype and stained positive or negative for UIS4 (*Figure 5A*, 2 and 44 hr, intracellular). UIS4$^+$ PVM is normally associated with a productive cell infection (*Mueller et al., 2005*). However, a small number of EEFs can develop in the absence of UIS (*Mueller et al., 2005*), likely inside the host cell nucleus (*Figure 5A*, 44 hr, intranuclear). In the control and 3D11-treated groups, the percentage of intracellular UIS4$^-$ parasites decreased two- to threefold from 2 to 44 hr, as expected of a parasite population negative for a marker of productive infection (*Figure 5B*). However, while at 2 hr in the control group, this population represented 14% of intracellular parasites, in the 3D11-treated group, it reached 48% (*Figure 5B*). This ~threefold increase in the UIS4-negative population could explain the late killing of intracellular sporozoites by 3D11. Whether this population is constituted by intracellular transmigratory sporozoites lacking a PVM or parasites surrounded by a PVM, but incapable of secreting UIS4 still needs to be determined. At 44 hr, surviving EEFs in the 3D11-treated samples presented a similar area and UIS4 staining intensity than control parasites (*Figure 5C, D*). However, as observed by flow cytometry (*Figure 4D*), the GFP intensity of 3D11-treated parasites was significantly lower than control EEFs, indicating that 3D11 can somehow affect protein expression with undetermined effects in the genesis of red blood cell infecting stages.

## Discussion

The skin plays a major role in parasite-dependent cytotoxicity induced by anti-CSP Abs (*Aguirre-Botero et al., 2023*; *Aliprandini et al., 2018*; *Flores-Garcia et al., 2018*; *Vanderberg and Frevert, 2004*). In the skin, mAbs inhibit sporozoite motility (*Aguirre-Botero et al., 2023*; *Aliprandini et al., 2018*; *Flores-Garcia et al., 2018*; *Vanderberg and Frevert, 2004*) and induce CSP shedding, leading to loss of their infectivity and death (*Aguirre-Botero et al., 2023*; *Aliprandini et al., 2018*). However, potent mAbs can also provide protection when parasites are introduced directly into the bloodstream

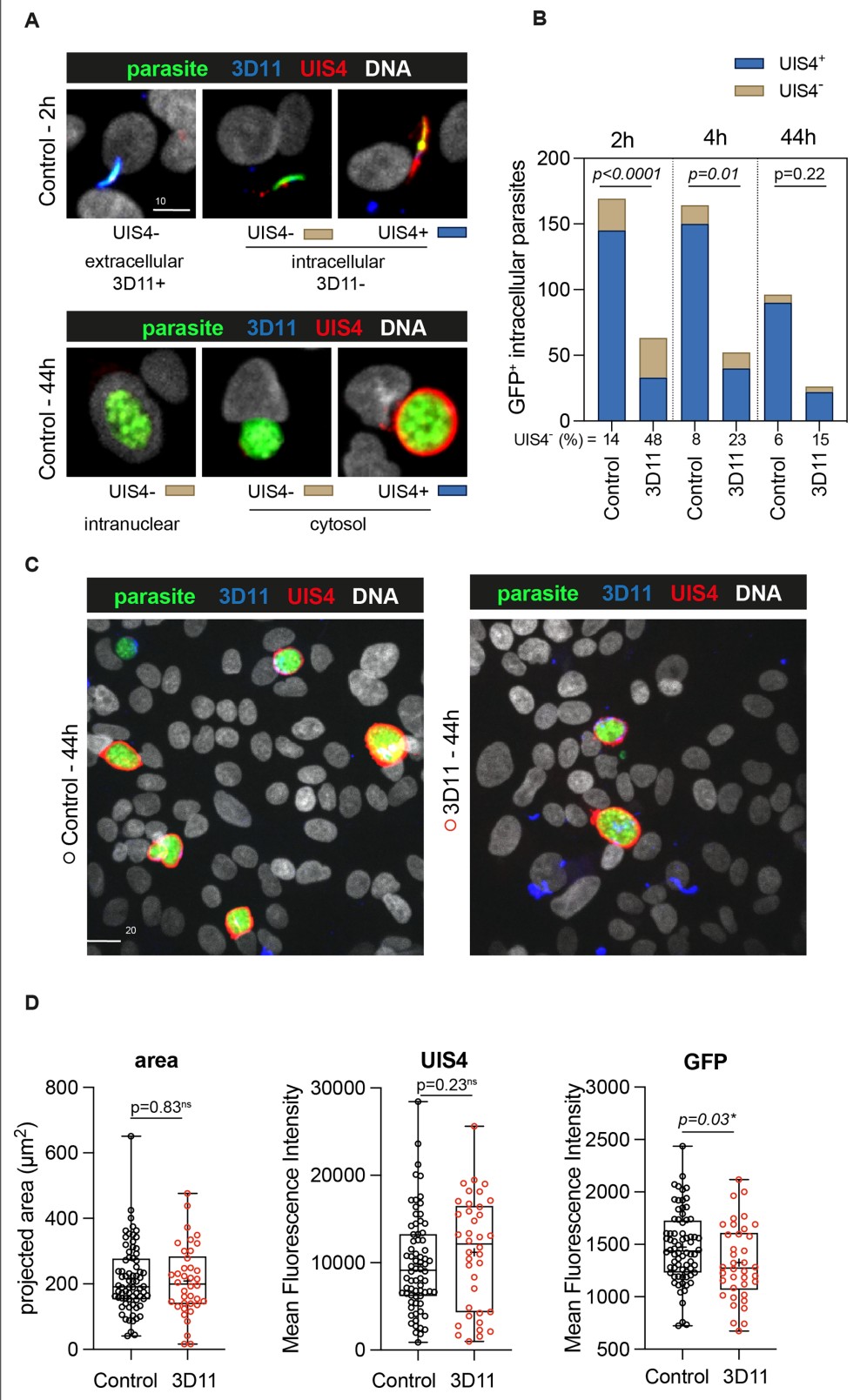

**Figure 5.** In vitro effect of 3D11 on intracellular parasites. HepG2 cells were incubated for 2 hr in the absence (Control) or presence (3D11) of 1.25 µg/ml 3D11 antibody. Cells were PFA fixed at 2 and 44 hr. Extracellular sporozoites were stained using 1 µg/ml of 3D11. UIS4 was labeled using polyclonal antibodies to assess the formation of the parasitophorous vacuole membrane (PVM). (**A**) Representative maximum projections of the

*Figure 5 continued on next page*

*Figure 5 continued*

double staining with 3D11 to differentiate between intra- and extracellular parasites and UIS4 at 2 and 44 hr post-sporozoite addition. Scale bar: 10 μm. (**B**) Quantification of the number of intracellular parasites with or without a positive UIS4 staining 2, 4, and 44 hr after sporozoites were added into the culture. (**C**) Representative images of intracellular parasites at 44 hr in the presence or absence of 3D11 for (**D**) comparison of their area, the mean fluorescence intensity of the UIS4, 3D11 staining, and GFP.

(*Aguirre-Botero et al., 2023*). Despite having lower protective efficiency in the BL than in the skin, this extra-cutaneous neutralizing activity is a characteristic of the most potent protective Abs (*Aguirre-Botero et al., 2023*), indicating that the killing of skin escapees is critical for sterilizing immunity. In the hepatic environment, these cytotoxic Abs have been shown to prevent parasite egress from sinusoids, hinder cell traversal, and kill sporozoites (*Wang et al., 2020*). However, the hepatic cytotoxic neutralizing activities of these different Abs do not always match their protective activity.

By combining intravital imaging and cellular assays, we quantified the *P. berghei* clearance by 3D11 focused on the different steps of parasite development in the liver. We first measured the effect of 3D11 in the BL and compared it to its effect in the skin by controlling the number of sporozoites for the i.v. and skin challenges. This allowed us to show that 3D11-mediated protection mainly unfolds in the skin, leading to sterile protection upon skin challenge, while there is only a tenfold decrease in parasitemia after i.v. challenge (*Figure 1A*). Thus, these data support the notion that 3D11 is less effective in the bloodstream (*Flores-Garcia et al., 2018*), as higher concentrations of Abs are required to protect against i.v. challenge compared to skin challenge (*Figure 1*). Accordingly, previous studies have reported that high amount of 3D11 transferred to mice is required to provide sterile protection after an i.v. challenge (*Charoenvit et al., 1991*; *Potocnjak et al., 1980*). To increase protection in the BL, instead of transferring 100 μg of 3D11 by i.p. injection and waiting 24 hr for the sporozoite inoculation, for the ensuing intravital imaging experiments, we transferred 100 μg of 3D11 by i.v. injection and inoculated parasites in the tail vein 10 min after the Ab transfer. This modification increased the protection in one order of magnitude (*Figure 2—figure supplements 1* and *Figure 3—figure supplement 1*).

When bioluminescent parasites are i.v. injected in passively immunized mice, 3D11 does not inhibit parasite arrest in the liver, suggesting that the repeat region of Pb CSP is not involved in the sporozoite homing to the liver. It is striking that once the parasites reach the liver, the bioluminescent signal in 3D11-treated mice only decreases ~30% at 4 hr post-sporozoite inoculation indicating that most parasites are lately killed by 3D11. A deeper and direct analysis of parasite elimination using intravital confocal microscopy showed that 3D11 mAb promptly inhibits parasite motility in the liver. After arresting in the liver in comparable numbers, ~80% of control PbPf sporozoites were motile, while only 40% of susceptible Pb mCherry sporozoites were moving (*Figure 3B*). However, the reduction in motility only prevented the hepatic parenchyma invasion of ~50% of parasites. These results suggest that 3D11 inhibits sporozoite motility and limits their exit from the liver sinusoids, as previously reported (*Wang et al., 2020*) but this twofold difference in parasite translocation alone cannot completely explain the ~100-fold reduction of parasitemia observed 3 days later in 3D11-treated versus control groups (*Figure 3—figure supplement 1B*). During 3 hr of continuous imaging, it was observed that 3D11-susceptible parasites rarely fragmented and died in vivo. These experiments were conducted in mice expressing GFP in endothelial cells, which made the accurate quantification of control PbPf GFP sporozoites difficult. Nonetheless, susceptible Pb mCherry parasites sporozoites were found in comparable numbers to the non-susceptible GFP sporozoites, indicating a low killing activity of 3D11 in the first 3 hr post-i.v. inoculation, and in concordance with the bioluminescence data. This observation contrasts with the findings on the effects of potent PfCSP human mAb (hmAbs) where 20 and 40% of sporozoites died while recording their in vivo effect in the liver (*Wang et al., 2020*). This discrepancy in parasite killing may be due to differences in Ab potency. However, this level of killing cannot also completely explain the magnitude of anti-CSP-mediated protection in both studies. To dissect the late-killing effect of the 3D11 mAb, we quantified the in vitro effect of this cytotoxic antibody in the process of invasion and sporozoite development inside hepatoma HepG2 cells. As previously described, 3D11 inhibits sporozoite cell invasion in vitro (*Hollingdale et al., 1982*). In our experimental setup, 5 μg/ml of 3D11 inhibited 88% of cell invasion (*Figure 4B, C*), which accounts for most of the 95% infection inhibition observed after 15 and 44 hr (*Figure 4C*). It is important to

note that the cytotoxicity of anti-CSP Abs correlates with sporozoite loss of fitness (*Aguirre-Botero et al., 2023*), quantified by the inhibition of motility, the loss of infectivity, and the killing of sporozoites (*Aguirre-Botero et al., 2023*; *Aliprandini et al., 2018*). The dose–response similarity between the inhibition of invasion and the decrease in viability, both measured in the same assay, suggests that the inhibitory activity of 3D11 on cell invasion assays could be a consequence of the cytotoxic effect of this antibody (*Figure 4B*). However, this does not rule out other mechanisms of invasion inhibition. Additionally, our data also show a lasting effect of 3D11 on intracellular parasites in vitro after Ab removal. This effect is reflected not only in the two- to threefold decrease in parasite numbers between 2 and 15 hr compared to the control using 2.5 and 5 µg/ml of 3D11 (*Figure 4D*) but also in the decrease of EEF fitness, revealed by its decreased GFP intensity (*Figures 4E and 5D*). Notably, a similar effect has been previously reported using sera from mice immunized with PfCSP or mAb against *Plasmodium yoelii* (Py) CSP. Incubation of Pf or Py sporozoites with the immune sera or mAbs not only affected sporozoite invasion in vitro but continued to affect intracellular forms for several days after invasion (*Mazier et al., 1986*; *Nudelman et al., 1989*). Additionally, using anti-Pf CSP sera, it was also observed that late EEFs from sera-treated sporozoites had abnormal morphology (*Mazier et al., 1986*). Altogether, it was thus concluded that the anti-CSP Abs present in the sera had a long-term effect on the parasites (*Mazier et al., 1986*; *Nudelman et al., 1989*). Importantly, in our experimental settings and using the mAb 3D11, we did not observe any malformation of the Pb EEFs but rather a decrease in the EEF GFP fluorescence at 44 hr in the 3D11-treated samples compared to the control. These differences could be due to the distinct invasion and developmental process of Pb and Pf, or to the potency of the Abs present in the serum compared to 3D11. The in vivo bioluminescence data suggest that 3D11 only weakly interferes with the development of late EEFs, as the signal of survivors in 3D11-treated mice between 24, 44, and 72 hr only slightly decreased, and this difference was not significative (*Figure 2D*, *Figure 2—figure supplement 1*). Accordingly, in vitro, the decrease on intracellular parasite numbers already occurred at 15 hr (*Figure 4C, F*) and stayed constant until 44 hr (*Figure 4C, D*). The GFP fluorescence decrease observed at 44 hr in vitro (*Figures 4E and 5C, D*), could explain the slight decrease in the bioluminescence signal observed in vivo between 24 and 72 hr (*Figure 2—figure supplement 1*). Finally, the relative percentual increase of the UIS4-negative population in 3D11-treated parasites versus control parasites indicates that the long-term killing could be a consequence of this phenotype. Further experiments are needed to determine whether the same phenotype is occurring in vivo and if the 3D11 lasting neutralizing activity is associated with the inhibition of UIS4 secretion or with the intracellular arrest of PVM-negative transmigratory parasites.

Although 3D11 cytotoxicity could be the primary contributor to the inhibition of parasite entry into the liver parenchyma in vivo and cell invasion in vitro, it is unclear whether it is the sole factor responsible for these effects. MAbs with high cytotoxicity, such as mAb10, do not significantly protect mice after i.v. sporozoite challenge, despite displaying a similar cytotoxic neutralizing activity profile in the liver than protective mAbs (*Aguirre-Botero et al., 2023*; *Wang et al., 2020*). CIS43, a hmAb that targets the junctional epitope of Pf CSP, has been found to interfere with its proteolytic cleavage (*Kisalu et al., 2018*), which is a crucial process for cell invasion (*Coppi et al., 2005*; *Espinosa et al., 2015*). Indeed, it has recently been shown using 13 distinct PfCSP hmAbs, that protection after i.v. challenge strongly correlates with high affinity to the junctional region, which contains all three repeat motifs of Pf CSP (NPDP, NVDP, and NANP) (*Aguirre-Botero et al., 2023*), reinforcing the hypothesis that anti-CSP repeat Abs could hinder the CSP processing at the putative cleavage site region I that is located adjacent to the junctional region of the central repetitive region of Pf CSP (*Coppi et al., 2005*). 3D11 Fab was shown to preferentially bind to the NPND motifs located mostly at the junction with the N-terminal domain. Still, it also cross-reacts with all repeat motifs, binding to PbCSP in a spiral-like conformation (*Kucharska et al., 2020*). Thus, further investigation is required to elucidate the role of CSP processing in inhibiting sporozoite invasion by anti-CSP Ab, for example, using parasites lacking the N-terminal domain of CSP (*Coppi et al., 2011*). To investigate other potential mechanisms by which Abs are neutralizing sporozoites in the BL, it is necessary to use non-cytotoxic conditions. This can be addressed by using a low-cytotoxic mAb, like CIS43, or parasites lacking sporozoite-secreted pore-forming proteins, such as SPECT2. It has been previously demonstrated that Ab-mediated killing depends on the shedding of CSP, which renders the parasite membrane susceptible to its own cell-wounding proteins (*Aliprandini et al., 2018*). According to this study, Pb and Py SPECT2KO mutants have a higher resistance to the killing activity of CSP Abs (*Aliprandini et al., 2018*).

In conclusion, this study shows the kinetics of sporozoite elimination by the cytotoxic 3D11 mAb in the liver and quantifies the inhibitory activity of 3D11 in the key steps of hepatic infection. The measurement of the inhibition of invasion and 3D11 cytotoxicity in vitro suggests a potential association between these two activities, but this result does not exclude the concomitant existence of other neutralizing mechanisms. Remarkably, this study reveals a late-killing effect of cytotoxic 3D11 on sporozoites within the liver parenchyma. This late-killing activity was also observed in vitro and it was associated with the increase of a population lacking the PVM developmental marker UIS4. This study shows that cytotoxic 3D11 mAb exerts its protective action outside the skin by inhibiting several critical steps in the hepatic infection and, surprisingly, taking several hours to significantly decrease the hepatic load of parasites. Understanding the mechanisms of sporozoite neutralization by anti-CSP Abs in the host tissues is crucial for better comprehending parasite biology and developing new tools to prevent sporozoite infection.

# Materials and methods

### Key resources table

| Reagent type (species) or resource | Designation | Source or reference | Identifiers | Additional information |
|---|---|---|---|---|
| Cell line (*H. sapiens*) | HepG2 | ATCC | HB-8065 | |
| Strain, strain background (*Mus musculus*, female) | RjOrl:SWISS | Janvier Laboratories | Cat#SN-SWISS-F | |
| Strain, strain background (*Mus musculus*, female) | C57BL/6JRj | Janvier Laboratories | Cat#SC-C57J-F | |
| Strain, strain background (*Anopheles stephensi*) | SDA 500 | CEPIA | N/A | |
| Strain, strain background (*Plasmodium berghei*) | ANKA, Pb GFP | *Ishino et al., 2006* | N/A | |
| Strain, strain background (*Plasmodium berghei*) | ANKA, Pb mCherry | *Aguirre-Botero et al., 2023* | N/A | |
| Strain, strain background (*Plasmodium berghei*) | ANKA, Pb GFP- luciferase | *Franke-Fayard et al., 2008* | N/A | |
| Strain, strain background (*Plasmodium berghei*) | ANKA, PbPf | *Wang et al., 2020* | N/A | |
| Antibody | 3D11 (mouse monoclonal) | BEI Resources | MRA-100 | (0.006–100 µg/ml) |

## Parasites, mosquitoes, and mice

For the i.v., skin, and bite challenges, we used transgenic Pb ANKA strain parasites expressing GFP (PbGFP) (*Ishino et al., 2006*). Pb parasites expressing GFP-luciferase (Pb GFP-luciferase) (*Franke-Fayard et al., 2008*) were used for bioluminescence imaging. For intravital and ex vivo microscopy of the infected livers, Pb ANKA expressing both GFP and the full-length Pf 3D7 CSP (PbPf GFP) (*Wang et al., 2020*) and Pb ANKA expressing mCherry (Pb mCherry) (*Aguirre-Botero et al., 2023*) were used.

*Anopheles stephensi* (strain SDA 500) mosquitoes were reared at the Center for Production and Infection of Anopheles (CEPIA) at the Institut Pasteur in Paris and infected by feeding for 30 min on infected RjOrl:Swiss mice. After blood feeding the mosquitoes were maintained at 21°C with 80% humidity. Sporozoites were obtained from the salivary glands of infected mosquitoes, isolated 20–26 days after infection.

Sporozoite challenge and bioluminescence experiments were performed on female 4-week-old C57BL/6JRj mice (Janvier Labs). For intravital and ex vivo experiments flk1-GFP female mice aged of 5–8 weeks were used. Mice were kept in the animal facility of the CEPIA at the Institut Pasteur in Paris accredited by the French Ministry of Agriculture for performing experiments on live rodents. All experiments were performed according to French and European regulations and were approved by the Ethics Committee #89 (references MESR 01324, APAFIS#32422-2021071317049057 v2, APAFIS #32989-2021091516594748 v1).

## mAb production and injection

The mouse 3D11 anti-Pb CSP repeats IgG1 mAb was produced by an established hybridoma cell line (MRA-100, BEI Resources) (*Eichinger et al., 1986*). W24 binds to the N-terminal domain of Py CSP and was used as an isotype control (*Aliprandini et al., 2018*). The hybridoma was cultured after the protocol of *Köhler and Milstein, 1975*. The mAbs were isolated from the culture supernatants by affinity chromatography using Protein G beads (GE Healthcare).

## Sporozoite challenge

Abs were diluted in 1× PBS and transferred intraperitoneally 24 hr before sporozoite injection or mosquito bite challenge. Mice were treated with the 3D11 mAb or, as control, with either PBS or W24 mAb. For the skin challenge, 5000 sporozoites were injected into the footpad of the mice using a 35 G needle mounted on a NanoFil 10 µl syringe (Word Precision Instruments). The i.v. challenge was performed by inoculating 1000 sporozoites in the tail vein. Bite challenge experiments were performed using infected female *A. stephensi* mosquitoes. Mice were anesthetized using a mixture of ketamine (60 mg/kg body weight; Imalgene 1000, Merial) and xylazine (6 mg/kg body weight; 2% Rompun, Bayern). Then mice were exposed to 10–16 mosquitoes (2 mice per pot) to be bitten for 30 min. The animals were rotated between the pots of each experimental group every 30–60 s.

To determine parasitemia, a drop of blood was collected at the tip of the tail in PBS 3–10 days after infection. Infected red blood cells were distinguished from non-infected by detecting the fluorescence emitted by the parasite's cytoplasmic GFP and quantified by cytometry (CytoFLEX S, Beckman). 250,000–500,000 events were recorded. Previously, the sample size was determined by comparing two unmatched group means using a parametric two-tailed test, taking experimental data into account.

## Bioluminescence experiments

Analysis of the parasite load within animals infected with Pb expressing luciferase sporozoites was performed as previously described (*Tavares et al., 2017*). Briefly, the abdominal areas of 4-week-old C57BL/6JRj female mice were shaved 24 hr before infection. Ten minutes after i.v. transfer of 100 µg of 3D11 Abs in 1× PBS, 150,000 sporozoites were i.v. injected. Immediately after parasite inoculation, mice were anesthetized with 2.5% isoflurane for 2 min. Subsequently, the animals were injected subcutaneously with 150 mg/kg body weight of D-luciferin/PBS (PerkinElmer). After 5 min the signal was acquired using the IVIS SpectrumCT (PerkinElmer). Succeeding acquisitions were taken 4, 24, and 44 hr later. For analysis, four ROIs were manually defined using the Living Image. For *Figure 2F*, the background noise was eliminated by subtracting from each ROI the signal obtained from the corresponding ROI applied to a non-infected mouse injected with D-luciferin.

## Intravital and ex vivo microscopy

For intravital imaging, the liver of flk1-GFP mice was surgically exposed by inserting on the abdomen of the animal a titanium ring covered by a 12-mm coverslip. Twenty-four hours after the operation, the mice were injected with 3D11 Ab, anesthetized, and positioned on the microscope stage where they were injected with $0.5 \times 10^6$ to $1 \times 10^6$ PbPf GFP and Pb mCherry parasites (1:1 ratio). Image acquisition was started immediately after parasite injection. The time-lapse between Ab and parasite injection was 10 min. For ex vivo experiments, no titanium window was inserted into the abdomen of the flk1-GFP mice. Ab and parasite injections were performed as described above in unanesthetized mice. One hour after the injections, the animals were anesthetized, the liver was surgically removed, and imaged.

For intravital and ex vivo microscopy experiments, livers were scanned systematically and for each field, a Z-stack and a time-lapse of 5–10 min were acquired over 2–3 hr after infection. Imaging was performed on a spinning-disk confocal system (UltraView ERS, PerkinElmer) controlled by Volocity (PerkinElmer) and composed of 4 Diode Pumped Solid State Lasers (excitation wavelengths: 405, 488, 561, and 640 nm), a Yokogawa Confocal Scanner Unit CSU22, a z-axis piezoelectric actuator and a Hamamatsu Orca-Flash 4.0 camera mounted on a Axiovert 200 microscope (Zeiss), using a 5–40× long-distance dry or oil-immersion objectives (Zeiss). Images were processed using Fiji (*Schindelin et al., 2012*).

## In vitro invasion and development assay

Sporozoite invasion and development of the EEFs were quantified by flow cytometry as previously described (*Prudêncio et al., 2008*). Briefly, HepG2 cells ($4 \times 10^4$/well) were seeded in a 96-well plate

and cultured overnight in DMEM (Invitrogen) supplemented with 10% heat-inactivated FCS, 1× of non-essential amino acids (Sigma) and 2% of penicillin–streptomycin–neomycin antibiotic mixture (Sigma). Uninfected and infected cells were maintained at 37°C and 5% $CO_2$. 10,000 PbGFP parasites were incubated with the cells (1:4 ratio) in the presence or absence of different 3D11 Ab concentrations. After 2 hr, the supernatant was collected, and the culture was washed 2× with 0.5 volume of PBS. The cells were subsequently trypsinized. The supernatant plus the washing steps and the trypsinized cells were analyzed by flow cytometry to quantify the amount of GFP+ events inside and outside cells (*Figure 4A*, *Figure 4—figure supplement 1*). Viability was then quantified by the sum of the total number of sporozoites (GPF+ events) in the supernatant, inside and outside the cells. We calculated the percentage of parasite viability by dividing the average of the total number of sporozoites in the treated samples by the average in controls using three technical replicates for each condition.

Additionally, we quantified the percentage of infected cells using the total number of GFP+ events in the HepG2 gate (*Figure 4—figure supplement 1*). To compare the biological replicates, we further normalized to the control of each experiment. For the samples used to analyze parasite development, the cells were incubated for 15 or 44 hr after sporozoite addition, and the medium was changed after 2 and 24 hr. The cells were trypsinized and the percentage of intracellular parasites was determined by flow cytometry as described above (*Figure 4—figure supplement 1*). The prolonged effect between 2 and 15/44 hr was calculated by normalizing the percentage of infected cells at 15/44 hr to that of 2 hr. For all flow cytometry measurements, the same volume was acquired.

## Immunofluorescence assays

Cells were infected and maintained for the immunofluorescence analysis of the parasites in the HepG2 culture, as described above. In this case, $2.5 \times 10^4$ cells per condition with 6250 sporozoites were incubated in a half 96-well plate with or without of 1.25 µg/ml 3D11. After 2, 4, and 44 hr, cultures were fixed with 4% formaldehyde. The double immunostaining protocol was used to differentiate between the extra- and intracellular parasites (*Rénia et al., 1988*; *Silvie et al., 2002*).

To visualize the presence of 3D11 in intracellular parasites, a goat anti-mouse Alexa 350 was first added before permeabilization to reveal extracellular parasites. After 0.2% Triton-X permeabilization, intracellular 3D11 was labeled with a donkey anti-mouse Alexa 647 nm. Parasites were imaged using an inverted AxioObserver microscope (Zeiss).

For the UIS4 staining, extracellular parasites were labeled with 1 µg/ml 3D11 and revealed with a goat anti-mouse Alexa 568 IgG. Cells were subsequently permeabilized, stained using a goat anti-UIS4 antibody, and revealed with a rabbit anti-goat Alexa 547 antibody. For nuclear visualization, 1 µg/ml Hoechst 33342 was used. Imaging was performed on a spinning-disk confocal system (UltraView ERS, PerkinElmer).

## Statistical analysis

Statistical significance was determined with a one-way ANOVA with Holm–Šídák correction for multiple comparisons, the Kruskal–Wallis test with Dunn's correction for multiple comparisons, paired and unpaired *t*-tests, and the Chi-square test with the help of GraphPad Prism 10.

## Acknowledgements

Special thanks to the team of the Center for Production and Infection of Anopheles (CEPIA, C2RA, Institut Pasteur) for providing mosquitoes and the staff of Central Animal Facility (C2RA, Institut Pasteur) for animal care and breeding. We gratefully acknowledge the UTechS Photonic BioImaging (Imagopole), C2RT, Institut Pasteur, supported by the French National Research Agency (France BioImaging; ANR-10-INBS-04; Investments for the Future Infrastructure en Biologie et Santé).

## Additional information

### Competing interests

Eduardo Aliprandini: Eduardo Aliprandini is affiliated with Astrazeneca. The author has no other competing interests to declare. The other authors declare that no competing interests exist.

## Funding

| Funder | Grant reference number | Author |
|---|---|---|
| Agence Nationale de la Recherche | ANR-19-CE15-0027 | Manuela C Aguirre-Botero<br>Olga Pacios<br>Susanna Celli<br>Eduardo Aliprandini<br>Anisha Gladston<br>Jean-Michel Thiberge<br>Pauline Formaglio<br>Rogerio Amino |
| Deutsche Forschungsgemeinschaft | ANR-19-CE15-0027 | Manuela C Aguirre-Botero<br>Olga Pacios<br>Susanna Celli<br>Eduardo Aliprandini<br>Anisha Gladston<br>Jean-Michel Thiberge<br>Pauline Formaglio<br>Rogerio Amino |
| Laboratoire d'Excellence Integrative Biology of Emerging Infectious Diseases | ANR-10-LABX-62-IBEID | Manuela C Aguirre-Botero<br>Olga Pacios<br>Susanna Celli<br>Eduardo Aliprandini<br>Anisha Gladston<br>Jean-Michel Thiberge<br>Pauline Formaglio<br>Rogerio Amino |
| Pasteur - Paris University (PPU) International | | Manuela C Aguirre-Botero |
| Institut Pasteur | | Manuela C Aguirre-Botero<br>Olga Pacios<br>Susanna Celli<br>Eduardo Aliprandini<br>Anisha Gladston<br>Jean-Michel Thiberge<br>Pauline Formaglio<br>Rogerio Amino |

The funders had no role in study design, data collection and interpretation, or the decision to submit the work for publication.

### Author contributions

Manuela C Aguirre-Botero, Conceptualization, Investigation, Writing – original draft, Writing – review and editing; Olga Pacios, Anisha Gladston, Jean-Michel Thiberge, Investigation; Susanna Celli, Investigation, Writing – review and editing; Eduardo Aliprandini, Supervision, Investigation; Pauline Formaglio, Supervision, Investigation, Writing – review and editing; Rogerio Amino, Conceptualization, Supervision, Funding acquisition, Writing – original draft, Writing – review and editing

### Author ORCIDs

Manuela C Aguirre-Botero ⓘ https://orcid.org/0000-0002-0019-5122
Olga Pacios ⓘ https://orcid.org/0000-0002-4476-856X
Susanna Celli ⓘ https://orcid.org/0000-0002-8296-895X
Pauline Formaglio ⓘ https://orcid.org/0000-0002-9612-3157
Rogerio Amino ⓘ https://orcid.org/0000-0002-8086-2932

### Ethics

All experiments were performed according to French and European regulations and were approved by the Ethics Committee #89 (references MESR 01324, APAFIS#32422-2021071317049057 v2, APAFIS #32989-2021091516594748 v1).

Reviewer #1 (Public review): https://doi.org/10.7554/eLife.105291.2.sa1
Reviewer #1 (Public review): https://doi.org/10.7554/eLife.105291.2.sa2
Reviewer #3 (Public review): https://doi.org/10.7554/eLife.105291.2.sa3
Author response https://doi.org/10.7554/eLife.105291.2.sa4

## Additional files

### Supplementary files
MDAR checklist

Source data 1. Raw data generated and presented in *Figures 1–5*.

### Data availability
All data generated or analyzed during this study are included in the manuscript and supporting file.

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
