## [Editor Report · eLife Assessment]

This **important** study shows that a monoclonal antibody against the repetitive region of the circumsporozoite protein (CSP) of the Malaria-causing parasite P. berghei has neutralizing activity on parasite invasion and development. The authors present **convincing** in vivo data confirming previous in vitro work, that suggested the intracellular post -invasion effect for this antibody. The findings offer insights into the inhibitory action of this anti-CSP antibody, which could inform the development of more effective malaria vaccines and therapeutic antibodies."

[Editors' note: this paper was reviewed by Review Commons.]

---

## [Referee Report · Reviewer #1 (Public review)]

The study by Aguirre-Botero et al. shows the dynamics of 3D11 anti-CSP monoclonal antibody (mAb) mediated elimination of rodent malaria Plasmodium berghei (Pb) parasites in the liver. The authors show that the anti-CSP mAb could protect against intravenous (i.v.) Pb sporozoite challenge along with the cutaneous challenge, but requires higher concentration of antibody. Importantly, the study shows that the anti-CSP mAb not only affects sporozoite motility, sinusoidal extravasation, and cell invasion but also partially impairs the intracellular development inside the liver parenchyma, indicating a late effect of this antibody during liver stage development. While the study is interesting and conducted well, the only novel yet very important observation made in this manuscript is the effect of the anti-CSP mAb on liver stage development.

Comments on latest version:

No further comments.

---

## [Referee Report · Reviewer #1 (Public review)]

The study by Aguirre-Botero et al. shows the dynamics of 3D11 anti-CSP monoclonal antibody (mAb) mediated elimination of rodent malaria Plasmodium berghei (Pb) parasites in the liver. The authors show that the anti-CSP mAb could protect against intravenous (i.v.) Pb sporozoite challenge along with the cutaneous challenge, but requires higher concentration of antibody. Importantly, the study shows that the anti-CSP mAb not only affects sporozoite motility, sinusoidal extravasation, and cell invasion but also partially impairs the intracellular development inside the liver parenchyma, indicating a late effect of this antibody during liver stage development. While the study is interesting and conducted well, the only novel yet very important observation made in this manuscript is the effect of the anti-CSP mAb on liver stage development.

Comments on latest version:

No further comments.

---

## [Referee Report · Reviewer #3 (Public review)]

Summary:

Aguirre-Botero et al have studied the effect of a potent monoclonal antibody against the circumsporozoite protein, the major surface protein of the malaria sporozoite. This is an elegantly designed, performed, and analyzed study. They have efficiently delineated the mode of action of anti-CSP repeat mAb and confirmed previous in vitro work (not cited) that demonstrated the same intracellular effect.

Major comments from the previous round of review:

Line 51: The authors claim a correlation between high antibody levels and protection. However, they did not provide direct proof that these antibodies were responsible for protection, nor did they establish a cut-off level of anti-CSP antibodies that would distinguish between protected and unprotected individuals.

Line 326: The late intrahepatic effect of mAb against the CSP repeat has been previously reported (see Figure 2, Nudelman et al, J Immunol, 1989). The effect was shown to affect the transition from liver trophozoites to liver schizonts. This study should be cited and discussed.

Significance:

A well-done study that elucidates the mechanisms of a protective monoclonal antibody against malaria sporozoites. These data are important and will interest a large audience of researchers working in infectious diseases and immunology.

Comments on latest version:

With the addition of new experiments and proper addition of missing references and minor text correction, the manuscript has been improved.

---

## [Author Response]

**Reviewer #1 (Evidence, reproducibility and clarity (Required)):**
The study by Aguirre-Botero et al. shows the dynamics of 3D11 anti-CSP monoclonal antibody (mAb) mediated elimination of rodent malaria Plasmodium berghei (Pb) parasites in the liver. The authors show that the anti-CSP mAb could protect against intravenous (i.v.) Pb sporozoite challenge along with the cutaneous challenge, but requires higher concentration of antibody. Importantly, the study shows that the anti-CSP mAb not only affects sporozoite motility, sinusoidal extravasation, and cell invasion but also partially impairs the intracellular development inside the liver parenchyma, indicating a late effect of this antibody during liver stage development. While the study is interesting and conducted well, the only novel yet very important observation made in this manuscript is the effect of the anti-CSP mAb on liver stage development.MajorThis observation is highlighted in the manuscript title but is supported by only limited data. A such it needs to be substantiated and a mechanism should be investigated. The phenomenon of intracellular effects of the anti-CSP mAb should be analyzed in much more detail. For example, can the authors demonstrate uptake of the Ab together with the parasite during hepatocyte invasion? What cellular mechanism leads to elimination?Lines 234 - 243; 308 - 325: These results are the gist of the entire study and also defined the title of the manuscript. Thus, it would be pre-mature to claim the substantial effect of 3D11 antibody in late killing of the parasite in the infected hepatocytes just by looking at the decreased GFP fluorescence. The authors need to at least verify the fitness of the liver stages by measuring the size of the developing parasites as well as using different parasite specific markers (UIS4, MSP1, HSP70 etc.) in immunofluorescence assays on the infected liver sections and in vitro infections.

We greatly appreciate the comments. We have taken the suggestions into consideration and deepened the characterization of 3D11's late killing of parasites. We first analyzed the presence of 3D11 in the intracellular parasite after the invasion and compared it with the CSP expression on the surface of control parasites (new Fig. 4F). Next, we tested a potential action of 3D11 added in the cell culture after the invasion (new Fig. 4G). The two new panels and the text accompanying them are shown below.

“Post-invasion labeling of 3D11 bound to the membrane of intracellular parasites revealed a strong staining surrounding the parasite at 2 and 15h, but only punctual traces of 3D11 at 44h (Figure 4F, 3D11, 3D11). Of note, CSP was detected surrounding the control parasites at all time-points indicating that the lack of staining at 44h is not due to a decrease in the CSP amount on the parasite surface (Figure 4F, CSP, Control). To evaluate the potential post-invasion entry of 3D11 into the PV of infected cells and posterior neutralization of intracellular parasites, we incubated invaded cells from 2 to 44 h with 3D11, but no effect on the parasite intracellular development was observed (Figure 4G, 2h p.i.). 3D11 incubated for 2 h with sporozoites and cells elicited, as expected, a dose-dependent inhibition of parasite development. Altogether, our results indicate that the late inhibition of parasite development is already achieved at 15h and likely caused by antibodies dragged inside cells bound to sporozoites before or during the invasion.”

Finally, we better characterized the parasite loss of fitness caused by 3D11 in infected cells by quantifying the parasite size, GFP intensity and the presence and intensity of UIS4, a parasitophorous vacuole membrane developmental marker at 2, 4 and 44h as described below in the new figure 5 and accompanying text.

“To further characterize the killing of intracellular parasites by 3D11 in HepG2 cells, we next evaluated the expression of the parasitophorous vacuole membrane (PVM) marker, UIS4 37, to infer the parasite intracellular development at 2, 4 and 44h. HepG2 cells were incubated with Pb-GFP expressing sporozoites in the absence (Control, Figure 5) or presence of 1.25 µg/mL of 3D11 during the first two hours of incubation (3D11, Figure 5). The chosen 3D11 concentration led to ~50% decrease in cell invasion (Figure 4C, 2h) and ~30% decrease in the post-invasion number of EEFs (Figure 4D), leaving enough parasites to be analyzed by microscopy. To distinguish between extracellular and intracellular parasites at 2h, washed and fixed samples were incubated with mouse 3D11 mAb (1µg/mL) and revealed with a fluorescent anti-mouse secondary antibody (Figure 5A, 3D11 in blue). Samples were then permeabilized and incubated with a goat anti-UIS4 polyclonal antibody revealed with a fluorescent anti-goat secondary antibody (Figure 5A, UIS4 in red). DNA was stained with Hoechst (Figure 5A, DNA in white).

Extracellular GFP+ sporozoites were identified by their 3D11+UIS4- phenotype (Figure 5,2h extracellular). Conversely, intracellular parasites were identified by their 3D11- phenotype and stained positive or negative for UIS4 (Figure 5, 2h and 44h, intracellular). UIS4+ PVM is normally associated with a productive cell infection 37. However, a small number of EEFs can develop in the absence of UIS4 37, likely inside the host cell nucleus (Figure 5, 44h, intranuclear).

In the control and 3D11-treated groups, the percentage of intracellular UIS4- parasites decreased 2 to 3-fold from 2 to 44h, as expected of a parasite population negative for a marker of productive infection (Figure 5B). However, while at 2h in the control group, this population represented 14% of intracellular parasites, in the 3D11-treated group, it reached 48% (Figure 5B). This ~3-fold increase in the UIS4 negative population could explain the late killing of intracellular sporozoites by 3D11. Whether this population is constituted by intracellular transmigratory sporozoites lacking a PVM or parasites surrounded by a PVM, but incapable of secreting UIS4 still needs to be determined. At 44h, surviving EEFs in the 3D11-treated samples presented a similar area and UIS4 staining intensity than control parasites (Figure 5C, D). However, as observed by flow cytometry (Figure 4D), the GFP intensity of 3D11-treated parasites was significantly lower than control EEFs, indicating that 3D11 can somehow affect protein expression with undetermined effects in the genesis of red blood cell infecting stages.”

Minor• Line 44 - 43: The statement is applicable only to the rodent infecting Plasmodium parasites. The authors need to clarify that.

This is an important clarification. We have modified the text that now reads:

“The sporozoite surface is covered by a dense coat of the circumsporozoite protein (CSP), shown to be an immunodominant protective antigen using a rodent malaria model”

• Line 68: Replace the second 'against' after the CSP with 'of'.

It is done.

• Line 141 - 143: The 3D11 mAb does affect the homing and killing in the blood of cutaneous injected sporozoites. The authors need to clearly state that the statement is true only for i.v. injected sporozoites.

Thank you for the comment. Now the text reads:

“Altogether, these data indicate that 3D11 rather than having an early effect on i.v. inoculated sporozoites in the blood circulation, e.g. by inhibiting the homing or killing the parasite in the blood, requires more than 4 h to eliminate most parasites in the liver.”

• Figure 3B: The numbers of sporozoites detected in the experiment varies from 0 h (line 172) to 2 h (line 184). Therefore, the numbers need to be mentioned on all the bars of each timepoint.

We have now added the numbers at the top of the graph from Figure 3B.

• Figure 3C: If the authors have used flk1-GFP mice, then how well they were able to detect the Pb-PfCSP GFP parasites in the vessel vs. parenchyma in the intravital imaging? The representative images for Pb-PfCSP GFP should also be included.

Since 3D11 does not target PbPf parasites most of them are motile in the movies, making them easily distinguishable from the endothelial cells. In addition, the stronger GFP intensity of sporozoites makes them detectable in the sinusoids. Representative images were added in the new Figure S3.

• It is not mentioned anywhere how the viability of the sporozoites was determined. This has to be described especially in the methods section.• Also, the flow acquisition and data analysis of the sporozoites and infected HepG2 cells must be described in the method section.

We briefly mentioned it in the results (line 228- 230): “In addition, by comparing the total number of recovered GFP+ sporozoites at 2 h in the two studied conditions, we measured the early lethality (%viable sporozoites, Figure 4B) of the anti-CSP Ab on the extracellular forms of the parasite (Figure 4A).”

A more detailed description has been added in the methods section that now reads:

“After 2 h, the supernatant was collected, and the culture was washed 2x with 0.5 volume of PBS. The cells were subsequently trypsinized. The supernatant plus the washing steps and the trypsinized cells were analyzed by flow cytometry to quantify the amount of GFP + events inside and outside cells (Figure 3A and Figure S4). Viability was then quantified by the sum of the total number of sporozoites (GPF + events) in the supernatant, inside and outside the cells. We calculated the percentage of parasite viability by dividing the average of the total number of sporozoites in the treated samples by the average in controls using three technical replicates for each condition. Additionally, we quantified the percentage of infected cells using the total number of GFP + events in the HepG2 gate (Figure S4). To compare the biological replicates, we further normalized to the control of each experiment. For the samples used to analyze parasite development, the cells were incubated for 15 or 44 h after sporozoite addition, and the medium was changed after 2 and 24 h. The cells were trypsinized and the percentage of intracellular parasites was determined by flow cytometry as described above (Figure S4). The prolonged effect between 2 h and 15/44 h was calculated by normalizing the percentage of infected cells at 15/44 h to that of 2 h. For all flow cytometry measurements, the same volume was acquired.”

• Figure 4: The flow layouts should be included for at least comparing the 0 vs. 5 μg/ml of 3D11 mAb concentrations.

Flow layouts were added in the supplementary figure 4.

• Line 651 (Figure S1 legend): Typographical error '14'.

Thank you for noticing. We corrected it.

**Reviewer #2 (Evidence, reproducibility and clarity (Required**)):Aguirre-Botero and collaborators report on the dynamics of Plasmodium parasite elimination in the liver using the 3D11 anti-CSP monoclonal antibody (mAb). By using microscopy and bioluminescence imaging in the P. berghei rodent malaria model, the authors first demonstrate that higher antibody concentrations are required for protection against intravenous sporozoite challenge, when compared to cutaneous challenge, which is not surprising. The study also shows that the 3D11 mAb reduces sporozoite motility, impairs hepatic sinusoidal barrier crossing, and more relevantly inhibits intracellular development of liver stages through its cytotoxic activity. These findings highlight the role of this specific monoclonal antibody, 3D11 mAb against CSP, in targeting sporozoites in the liver.Major CommentsThe study provides valuable insights into the mechanisms of protection conferred by the 3D11 anti-CSP monoclonal antibody against P. berghei sporozoites and this finding allow the field to speculate that other monoclonal antibodies against CSP of *P. falciparum* may act similarly. However, an important experiment is missing that would significantly strengthen the conclusions. Specifically, the authors should perform experiments where the monoclonal antibody is added immediately after the sporozoites have completed invasion. This should be done both in vitro and in vivo to show whether the antibody has any effect on intracellular development of liver stages when added after invasion.While the claims are generally supported by the data presented, to comprehensively conclude the late cytotoxic effects of 3D11, the additional experiment of post-invasion antibody application is relevant. This would help determine if the observed effects are due to the antibody's action during invasion or its continued action post-invasion.The data and methods are presented in a manner that allows for reproducibility. The use of microscopy and bioluminescence imaging is well-documented. The experiments appear adequately replicated, and statistical analyses are appropriate.

We thank reviewer 2 for these important suggestions. To be sure that the effect might not come from the internalization of the antibodies after sporozoite invasion, we tested the amount of 3D11 bound to the parasite following invasion (new Fig. 4F) and the potential post-invasion neutralizing effect of 3D11 in vitro. The results obtained are presented below.

“Post-invasion labeling of 3D11 bound to the membrane of intracellular parasites revealed a strong staining surrounding the parasite at 2 and 15h, but only punctual traces of 3D11 at 44h (Figure 4F, 3D11, 3D11). Of note, CSP was detected surrounding the control parasites at all time-points indicating that the lack of staining at 44h is not due to a decrease in the CSP amount on the parasite surface (Figure 4F, CSP, Control). To evaluate the potential post-invasion entry of 3D11 into the PV of infected cells and posterior neutralization of intracellular parasites, we incubated invaded cells from 2 to 44 h with 3D11, but no effect on the parasite intracellular development was observed (Figure 4G, 2h p.i.). 3D11 incubated for 2 h with sporozoites and cells elicited, as expected, a dose-dependent inhibition of parasite development. Altogether, our results indicate that the late inhibition of parasite development is already achieved at 15h and likely caused by antibodies dragged inside cells bound to sporozoites before or during the invasion.”

Minor CommentsThe text and figures are clear and accurate. Some minor typographical errors should be corrected.

Thank you for the remark; we have verified the text again to remove typographical errors.

**Reviewer #3 (Evidence, reproducibility and clarity (Required)):**
Aguirre-Botero et al have studied the effect of a potent monoclonal antibody against the circumsporozoite protein, the major surface protein of the malaria sporozoite. This is an elegantly designed, performed, and analyzed study. They have efficiently delineated the mode of action of anti-CSP repeat mAb and confirmed previous in vitro work (not cited) that demonstrated the same intracellular effect.Specific commentsLine 51: The authors claim a correlation between high antibody levels and protection. However, they did not provide direct proof that these antibodies were responsible for protection, nor did they establish a cut-off level of anti-CSP antibodies that would distinguish between protected and unprotected individuals.

We thank reviewer 3 for the comments. Indeed, we agree with reviewer 3, these are correlative studies where the causality cannot be established. We modified the ensuing sentence to specify the causality between anti-CSP mAbs and in vivo protection against sporozoite infection. Now the text reads:

“Extensive research has demonstrated a positive correlation between high levels of anti-CSP antibodies (Abs) induced by the RTS,S/AS01 vaccine and efficacy against malaria(11-13). Remarkably, anti-CSP monoclonal Abs (mAbs) have been proven to protect *in vivo* against malaria in various experimental settings, including, mice(14-21), monkeys(23), and humans(24-26)”

Line 326: The late intrahepatic effect of mAb against the CSP repeat has been previously reported (see Figure 2, Nudelman et al, J Immunol, 1989). The effect was shown to affect the transition from liver trophozoites to liver schizonts. This study should be cited and discussed.

Thank you for this important remark. We included this seminal reference and now the modified text reads:

“Notably, a similar effect has been previously reported using sera from mice immunized with PfCSP or mAb against P. yoelii (Py) CSP. Incubation of Pf or Py sporozoites with the immune sera or mAbs not only affected sporozoite invasion in vitro but continued to affect intracellular forms for several days after invasion(38,39). Additionally, using anti-PfCSP sera, it was also observed that late EEFs from sera-treated sporozoites had abnormal morphology(38). Altogether, it was thus concluded that the anti-CSP Abs present in the sera had a long-term effect on the parasites(38,39).”